# A Simple Non-Invasive Score Based on Baseline Parameters Can Predict Outcome in Patients with COVID-19

**DOI:** 10.3390/vaccines10122043

**Published:** 2022-11-29

**Authors:** Riccardo Scotto, Amedeo Lanzardo, Antonio Riccardo Buonomo, Biagio Pinchera, Letizia Cattaneo, Alessia Sardanelli, Simona Mercinelli, Giulio Viceconte, Alessandro Perrella, Vincenzo Esposito, Alessio Vinicio Codella, Paolo Maggi, Emanuela Zappulo, Riccardo Villari, Maria Foggia, Ivan Gentile

**Affiliations:** 1Department of Clinical Medicine and Surgery—Section of Infectious Diseases, University of Naples Federico II, 80131 Naples, Italy; 2Emerging Infectous Disease with High Contagiousness Unit, Cotugno Hospital, 80131 Naples, Italy; 3IVth Division of Immunodeficiency and Gender Infectious Diseases, Cotugno Hospital, 80131 Naples, Italy; 4Department of Medical Sciences—Unit of Infectious Diseases, "Gaetano Rummo” Hospital, 82100 Benevento, Italy; 5Infectious and Tropical Diseases Clinic, AORN Sant'Anna and San Sebastiano, 81100 Caserta, Italy

**Keywords:** COVID-19, SARS-CoV-2, C-reactive protein, P/F ratio, LDH, mortality

## Abstract

We evaluated the role of CRP and other laboratory parameters in predicting the worsening of clinical conditions during hospitalization, ICU admission, and fatal outcome among patients with COVID-19. Consecutive adult inpatients with SARS-CoV-2 infection and respiratory symptoms treated in three different COVID centres were enrolled, and they were tested for laboratory parameters within 48 h from admission. Three-hundred ninety patients were enrolled. Age, baseline CRP, and LDH were associated with a P/F ratio < 200 during hospitalization. Male gender and CRP > 60 mg/L were shown to be independently associated with ICU admission. Lymphocytes < 1000 cell/μL were associated with the worst P/F ratio. CRP > 60 mg/L predicted exitus. We subsequently devised an 11-points numeric ordinary scoring system based on age, sex, CRP, and LDH at admission (ASCL score). Patients with an ASCL score of 0 or 2 were shown to be protected against a P/F ratio < 200, while patients with an ASCL score of 6 to 8 were shown to be at risk for P/F ratio < 200. Patients with an ASCL score ≥ 7 had a significantly increased probability of death during hospitalization. In conclusion, patients with elevated CRP and LDH and an ASCL score > 6 at admission should be prioritized for careful respiratory function monitoring and early treatment to prevent a progression of the disease.

## 1. Introduction

The COVID-19 pandemic is a serious threat to global health, with an impact that is unprecedented within the last 100 years. On 8 June 2022, there were 530,896,347 confirmed cases of COVID-19, including 6,301,020 deaths reported to the WHO [1]. After 2 years of the pandemic, the efforts in the research moved towards the development of vaccines with excellent effectiveness and safety profiles for the prevention of COVID-19 [2,3,4]. The availability of both mRNA-based and viral vector vaccines significantly succeeded in minimizing the burden of the pandemic worldwide in terms of severe disease progression, hospitalizations, and deaths [5,6]. Nevertheless, there are sub-groups of patients who remain at high risk for severe disease, intensive care unit (ICU) need, and death, regardless of vaccination status. [7,8,9,10]. It is indeed well-known that older age, obesity, chronic comorbidities (e.g., diabetes mellitus, cardiovascular and pulmonary diseases, chronic liver and kidney diseases), and immunodeficiency can put patients at high risk for severe COVID-19 [11]. These patients can easily progress to the hyperinflammatory phase of SARS-CoV-2 infection [12]. At this stage, markers of systemic inflammation are elevated, and SARS-CoV-2 infection can result in a decrease in helper, suppressor, and regulatory T cell counts [13]. Studies have shown that inflammatory cytokines and biomarkers such as IL-2, IL-6, IL-7, granulocyte colony-stimulating factor, macrophage inflammatory protein 1-a, tumour necrosis factor-a, CRP, ferritin, and d-dimer are significantly elevated in those patients with more severe disease [12,14,15]. A meta-analysis showed that elevated CRP, elevated LDH, and lymphopenia were among the most prevalent laboratory findings in patients with COVID-19 [16]. In the same study, a correlation between these laboratory abnormalities and the severity of COVID-19 was found. Namely, patients with increased CRP, increased LDH, or lymphopenia were found to be at high risk for severe COVID-19. However, a direct correlation between CRP and LDH levels in COVID-19 patients and the worsening of the respiratory function has not been proven yet. Given such considerations, the aim of the present study was to analyse the correlation between laboratory parameters at admission (including CRP) in patients hospitalized for COVID-19 and the rate of deterioration of the respiratory function after admission.

## 2. Material and Methods

### 2.1. Study Design

A multicentre retrospective study was conducted among adult inpatients with COVID-19 hospitalized between January 2020 and April 2021 and referring to the following clinical centres:-Unit of Infectious Diseases. University Hospital Federico II, Naples.-Hospital “D. Cotugno”. AORN “Dei Colli”, Naples.-Hospital “G. Rummo”, Benevento.-Hospital “Sant’Anna e San Sebastiano”, Caserta.

All included patients had a diagnosis of SARS-CoV-2 infection performed with a molecular (PCR) nasal and oropharyngeal swab and were hospitalized for COVID-19-related symptoms. The following exclusion criteria were applied:-Absence of respiratory symptoms related to COVID-19.-No serum CRP performed at admission (within 48 h).-No serum LDH performed at admission (within 48 h).-No arterial blood gas (ABG) test performed at admission (within 48 h).-History of a previous SARS-CoV-2 infection or presence of positive SARS-CoV-2 molecular test antecedent 2 weeks from hospitalization.-History of SARS-CoV-2 vaccination.-Other hospitalizations in the previous 30 days.

Respiratory symptoms related to COVID-19 included: cough, dyspnoea, tachypnoea, and respiratory failure. Extra-pulmonary manifestations of COVID-19 were not considered for inclusion in the present study.

The primary objective of the study was to analyse the correlation between serum CRP at hospital admission and the worst partial pressure of arterial oxygen to fraction of inspired oxygen ratio (P/F ratio) observed during hospitalization in patients with COVID-19-related respiratory symptoms. Secondary objectives were:-To analyse the correlation between serum LDH at hospital admission and the worst P/F ratio observed during hospitalization.-To analyse the correlation between blood lymphocyte count at admission and the worst P/F ratio observed during hospitalization.-To analyse the presence of risk factors for the worst P/F ratio < 200 during hospitalization.-To elaborate a scoring system for prediction of respiratory function deterioration.-To investigate the presence of risk factors for intensive care need during hospitalization.-To investigate the presence of risk factors for death during hospitalization.

The clinical records of all included patients were revised, and the following data were collected and reported in an electronic dataset: demographic and clinical data, main comorbidities, laboratory parameters (including CRP, LDH, white blood count), ABGs, and outcomes (ICU need and death). All laboratory parameters were collected at admission (within 48 h) and every 7 days from admission. All results from ABGs performed during hospitalization were collected, and the P/F ratios were calculated. The lowest value of P/F ratio observed during hospitalization for each patient were collected and reported as “worst P/F ratio”.

The study was conducted according to the guidelines of the Declaration of Helsinki and approved by the Ethics Committee of University of Naples Federico II (Protocol number 98/2022; Sperimentation I.D. 1032)

### 2.2. Statistical Analysis

All the variables were tested for parametric/non-parametric distribution with the Kolmogorov–Smirnov test. Comparisons between categorical dichotomous variables were performed with the χ^2^ test (or with Fischer’s exact test when applicable), while comparisons between ordinary variables were conducted with the Student T test (parametric variables) or the Mann–Whitney U test (non-parametric variables). Comparisons of demographic and laboratory parameters were stratified according to three different clinical outcomes: rate of patients with worst P/F ratio < 200 (meant as the lowest P/F ratio observed during the entire hospitalization for each patient), ICU admission during hospitalization, and death. The Spearman’s test and the linear regression analysis were used to correlate demographic (age) and laboratory parameters (CRP, LDH, lymphocyte count) with ordinary clinical parameters (namely, worst P/F ratio during hospitalization). The multivariate linear regression analysis was performed, including all the parameters significantly correlated with the dependent variable in the univariate linear regression analysis with a *p* < 0.2. In order to identify independent predictors for the three clinical outcomes (worst P/F ratio <200 during hospitalization, ICU admission, death), a logistic regression model was used. Parameters associated with the dependent variables (*p* < 0.2) in the univariate analysis were then included in a multivariate model. A predictive score for worst P/F ratio < 200 during hospitalization was elaborated according to the results of the logistic regression analysis. The age of patients was included in the predictive score, based on which it was categorized using the same cut-offs as Charlson’s comorbidity index [17]. The predictive score was correlated with the worst P/F ratio during hospitalization using the Spearman’s test, logistic regression analysis (ordinary worst P/F ratio), and logistic regression analysis (worst P/F ratio < 200). The diagnostic accuracy for the worst P/F ratio < 200 of the predictive score was evaluated with ROC curves. For all the tests, a *p*-value < 0.05 was considered significant. IBM SPSS© version 27 was used for statistical analysis.

## 3. Results

Globally, 323 patients from the four participant centres were included in the study in accordance with the inclusion/exclusion criteria. Demographic and clinical characteristics of the included patients are reported in Table 1.

Most patients showed impaired laboratory parameters within 48 h from hospital admission, with 35.9% and 50.8% of patients showing CRP values above 60 mg/L and a lymphocyte count below 1000 cell/µL, respectively. Only a minority of patients (4.6%) showed LDH values above 600 IU/L, but 142 patients (44%) had LDH values above 300 IU/L within 48 h from hospital admission. The median worst P/F ratio observed during hospitalization was 207 (IQR: 124–301) and nearly half of all the included patients (47.4%) had a worst P/F ratio below 200 during hospitalization. Rate of ICU admission and death rate were 15.8% and 6.8%, respectively. The differences in demographic, clinical, and laboratory parameters according to the presence of the three unfavourable outcomes are shown in Table 2. 

Patients with a worst P/F ratio below 200 during hospitalization were more frequently male (*p* < 0.05) and older (*p* < 0.001) than those with P/F ≥ 200. Patients who needed ICU admission were more frequently male compared to those with no ICU necessity (*p* < 0.001), while those who had a fatal outcome were older than those who survived (*p* < 0.001). Baseline CRP levels were found to be significantly higher among patients with a worst P/F ratio < 200 during hospitalization (*p* < 0.001) and those who died (*p* < 0.001). Baseline LDH levels were also higher among patients with a worst P/F ratio < 200 during hospitalization (*p* < 0.001) and those with a fatal outcome (*p* < 0.05). LDH levels were also found to be higher among patients who needed ICU admission (*p* < 0.001). Finally, blood lymphocyte count at admission was lower among patients with a worst P/F ratio during hospitalization (*p* < 0.001), those who needed ICU admission (*p* < 0.05), and those who died (*p* < 0.01). Interestingly, the presence of comorbidities was not associated with a worst P/F ratio < 200 nor with ICU admission. However, among patients who survived, most had no comorbidities (*p* < 0.001), while patients who had a fatal outcome more frequently had 1–2 comorbidities (*p* < 0.05) or 3–5 comorbidities (*p* < 0.01) compared to those who survived. When literature-derived cut-offs for CRP, LDH, and lymphocyte blood count were applied, it was found that CRP > 60 mg/L, LDH > 600 IU/L, and lymphocyte < 1000 cell/µL were associated with all three unfavourable outcomes (worst P/F ratio < 200 during hospitalization, ICU admission, death). Given the paucity of patients with LDH levels above 600 IU/I, a cut-off of 300 IU/L was also applied. The prevalence of patients with LDH > 300 was higher among patients with a worst P/F ratio below 200 during hospitalization (*p* < 0.001) and those who were admitted to the ICU (*p* < 0.05), compared with patients with a worst P/F ratio above 200 and those who did not need ICU admission, respectively. No differences in the rate of patients with LDH > 300 IU/L were found among patients who had a fatal outcome when compared with those who survived. 

In the correlation analysis, a significant and inverse correlation was found between the worst P/F ratio during hospitalization and the following variables: age (Spearman’s r = −0.299, *p* < 0.001); basal CRP values (Spearman’s r = −0.293, *p* < 0.001); basal LDH values (Spearman’s r = −0.363, *p* < 0.001). On the other hand, a direct correlation was found between the worst P/F ratio during hospitalization and basal lymphocyte count (Spearman’s r = 0.250, *p* < 0.001). In the linear regression analysis, a significant and negative association was found between the worst P/F ratio during hospitalization (dependent variable) and the following variables: age (B = −2.372, r^2^ = 0.125, *p* < 0.001); baseline CRP (B = −0.504, r^2^ = 0.084, *p* < 0.001, Figure 1); baseline LDH (B = −0.256, r^2^ = 0.116, *p* < 0.001, Figure 2). 

The lymphocyte count at admission was not significantly associated with the worst P/F ratio during hospitalization in the regression analysis.

Interestingly, age, baseline CRP values, and baseline LDH values were significantly associated with the worst P/F ratio during hospitalization in the multivariate linear regression analysis (all *p* < 0.001) (Table 3).

The results of the logistic regression analysis are shown in Table 4. Male sex (aOR: 1.73, *p* < 0.05), age > 60 years (aOR: 1.80, *p* < 0.05), CRP > 60 mg/L (aOR: 2.33, *p* < 0.01), and LDH > 300 IU/L (aOR: 2.47, *p* < 0.001) were shown to be independently associated with a worst P/F ratio below 200 during hospitalization. Male sex (aOR: 2.31, *p* < 0.05) and CRP > 60 mg/L at admission (aOR: 2.00, *p* < 0.05) were shown to be independently associated with ICU admission in the multivariate analysis. Finally, age > 60 years (aOR: 8.65, *p* < 0.01), the presence of 3–5 chronic comorbidities (aOR: 8.17, *p* < 0.01), and CRP > 60 mg/L at admission (aOR: 5.45, *p* < 0.01) were independently associated with death.

Given the results from the linear and logistic regression analysis for worst P/F ratio < 200, an 11-points numeric ordinary scoring system based on age, sex, CRP at admission, and LDH at admission (ASCL score) was elaborated (Table 5). 

The median ASCL score among patients included in the study was 5 (IQR: 3–6). The higher the ASCL score, the higher the risk was for P/F < 200 during hospitalization (Table 6). In particular, patients with an ASCL score of 0 (OR: 0.20; 95CI: 0.07 to 0.60) or 2 (OR: 0.43; 95CI: 0.20 to 0.90) were shown to be protected against a P/F ratio < 200, while patients with an ASCL score of 6 (OR: 2.31; 95CI: 1.18 to 4.52), 7 (OR: 3.30; 95CI: 1.35 to 8.09), or 8 (OR: 2.54; 95CI: 1.16 to 5.59) were shown to be at risk for P/F ratio < 200. No patients with an ASCL score of 1(*n* = 4) had a P/F < 200, while all the patients with an ASCL score of 10 (*n* = 5) showed a worst P/F ratio < 200 during hospitalization. 

The regression analysis for the worst P/F ratio (dependent variable) showed that for each 1-point increase in the ASCL score, a reduction in the worst P/F ratio of approximately 22 is expected (B = −21.65; 95CI: −26.16 to −17.15, r^2^ = 0.223, *p* < 0.001) (Figure 3). 

The diagnostic accuracy of the ASCL score for P/F ratio deterioration below 200 was fair (AUC: 0.717, *p* < 0.001) (Figure 4). 

Finally, the ASCL score was significantly higher among patients who died (median: 7; IQR: 6–8) compared with patients who survived (median: 4; IQR: 3–6, *p* < 0.001). The diagnostic accuracy of the ASCL score for death was good (AUC: 0.804, *p* < 0.001). Patients with an ASCL score ≥ 7 had a significantly higher probability of death during hospitalization (OR: 6.37; 95CI: 2.59 to 15.65, *p* < 0.001) than those with an ASCL less than 7.

## 4. Discussion

The pandemic of coronavirus disease 2019 (COVID-19) has caused an unprecedented global, social, and economic impact and high numbers of deaths. The clinical features of COVID-19 are diverse and range from asymptomatic presence to critical illness and death, with severe and critical cases represented by 14% and 5% of laboratory-confirmed COVID-19 patients, respectively [18]. A good understanding of the possible risk factors in combination with disease immunopathology associated with COVID-19 severity is helpful for clinicians in identifying patients who are at high risk and require prioritized treatment to prevent disease progression and adverse outcomes [19]. Risk factors range from demographic factors, such as age [20,21,22], sex and ethnicity [23,24], diet, and lifestyle habits [25,26], to underlying diseases [27,28,29,30,31,32] and complications [33,34,35,36]. Several laboratory abnormalities were also associated with increased risk of severe COVID-19 and disease progression [37,38,39,40,41,42,43,44,45]. In particular, according to the “Rule-of-6” by Dickens BSL et al., the presence with SARS-CoV-2 infection within 48 h from hospital admission of CRP > 60 mg/L, ferritin > 600 µg/L, and LDH > 600 IU/L aided in early identification of COVID-19 patients at risk of deterioration to the point of ICU admission [46]. In our study, we found that patients with respiratory deterioration had higher levels of CRP and LDH and a lower lymphocyte count compared with patients with a P/F > 200 during hospitalization. Similar results were obtained comparing laboratory parameters in patients requiring or not requiring ICU admission and in patients who survived compared with those with a fatal outcome, as described in other studies [47,48,49]. Interestingly, in our study, the presence of comorbidities was not associated with a P/F < 200 or with ICU admission. However, cardiovascular disease, CKD, malignancy, and diabetes, as well as the presence of at least one comorbidity, were significantly more frequent in patients with a fatal outcome. This result indicates that, in contrast to laboratory parameters and other demographic characteristics (e.g., age), the presence of comorbidities does not directly influence respiratory function and mechanics. In fact, we found an inverse and significant association between age (*p* < 0.001) serum CRP (*p* < 0.001, Figure 1) and LDH (*p* < 0.001, Figure 2) and the values of the worst P/F ratio during hospitalization in the multivariate analysis (Table 3). Moreover, male sex, age > 60 years, CRP > 60 mg/L, and LDH > 300 IU/L were independently associated with respiratory deterioration (P/F below 200 during hospitalization). CRP > 60 mg/L was found to be an independent risk factor for ICU admission and death, while LDH > 300 IU/L only showed an association with ICU admission in the univariate logistic regression analysis. The blood lymphocyte count < 1000 cells/µL was not associated with P/F < 200, probably because the white blood count of patients with COVID-19 is considerably influenced by the inflammatory status and, thus, dependant on CRP values. Similarly, no associations were found between lymphocyte < 1000 cells/µL and ICU admission or death. Finally, the presence of at least three comorbidities was found to be an independent risk only for mortality. In accordance with the results from the multivariate logistic regression analysis for a P/F < 200 during hospitalization, we devised a score based on age, sex, CRP and LDH (ASCL score, Table 6). A progressive increase in the ASCL score was found to be significantly associated with disease progression and respiratory function deterioration, as well as with the risk of death.

There is a spate of literature showing the association between lymphopenia and COVID-19 severity [47,50]. The decreased lymphocyte counts might be caused by viral attachment, immune injuries from inflammatory mediators, or exudation of circulating lymphocytes into inflammatory lung tissues [51].

Elevated serum LDH levels have been widely reported in COVID-19 cases and were predominantly higher in severe patients [52]. A meta-analysis showed that the mean value of LDH in severe patients with COVID-19 was 1.54 times higher than in non-severe cases [53]. The positive correlation between levels of LDH and disease severity makes it a valuable candidate biomarker for monitoring severe COVID-19 patients. Since higher levels of LDH have been observed in non-survivors at the early stage of illness [48], measuring this parameter at admission will be of greater predictive value for patients’ death risk. Elevated LDH values were shown to be correlated with the lung injury Murray score in patients with COVID-19 [54], and thus, elevated LDH values at the early stages of SARS-CoV-2 infection can likely predict a severe deterioration of respiratory function. 

Finally, elevated CRP is a key marker of disease progression and a risk factor for mortality in COVID-19 patients, and it is indicative of developing a cytokine storm in COVID-19 patients [20,55]. Out of 32 studies, 20 showed a nearly four-fold higher risk of poor outcomes in COVID-19 patients with elevated CRP [49]. Moreover, analysis of patients admitted to the ICU showed an increase in CRP levels in the first seven days [56], suggesting that CRP levels may be correlated with lung injury and respiratory function in patients with COVID-19. Although the role of CRP as a predictive factor for disease progression and mortality in patients with SARS-CoV-2 infection has been widely established, a direct correlation between CRP levels and respiratory function is yet to be documented. As an indicator of a triggered cytokine storm, elevated CRP levels in the early phases of the infection may predict subsequent lung damage and respiratory function deterioration caused by the hyper-inflammatory status in patients with COVID-19, as is conceivable from the results of Dickens et al. [46] Aside from the limited number of patients included in their cohort, the scoring system by Dickens et al. did not consider the prognostic weight of demographic factors (such as sex, age, and ethnicity), which are known to heavily influence the prognosis of patients with COVID-19. Finally, the authors did not clarify whether their score was correlated with respiratory function deterioration or other clinical variables. 

We acknowledge that this study had some limitations, especially in consideration of its retrospective nature, which partially compromised the data collection. In fact, several patients were excluded from the study, as they did not perform CRP/LDH or ABG at admission. Only a minority of patients needed ICU admission (51, 15.8%) or had a fatal outcome (22, 6.8%), and this must be taken into account in interpreting the results. Nevertheless, we believe that the sample size was sufficient to draw significant conclusions regarding correlations with the worst P/F. Moreover, it is known that serum CRP levels in patients with COVID-19 may be affected by the presence of bacterial co-infections. In this cohort of patients, the presence of bacterial co-infections at admission was not systematically evaluated. However, a systematic review and meta-analysis showed that the co-infection rate among patients with COVID-19 is relatively low (7%) [57], and this rate is even lower when considering the presence of bacterial co-infections at hospital admission (3%) [58]. Having said that, a routine and systematic screening for bacterial infections at hospital admission in patients with COVID-19 is not recommended, and we believe that the possible presence of bacterial co-infections at hospital admission among patients included in this study cohort is unlikely to have affected the results. Finally, the ASCL score must be validated in more numerous prospective cohorts to draw significant conclusions regarding its diagnostic accuracy in predicting respiratory deterioration in patients with COVID-19. Despite the discussed limitations, we showed a correlation between parameters at admission and the further worsening of respiratory function in patients with COVID-19. Such evidence supports the use of only a few parameters collected early in hospitalization to predict progression of the disease.

In conclusion, despite the above-mentioned limitations, the results from this study showed that CRP and LDH levels at admission correlate well with the deterioration of respiratory function in patients with COVID-19. A score based on age, sex, CRP and LDH at admission seems to have a good predictive role in the progression of the respiratory clinical picture and in identifying patients at high risk for unfavourable outcome. Patients with CRP > 60 mg/L or LDH > 300 IU/L at hospital admission, as well as patients with an ASCL score > 6 at hospital admission, should be prioritized for careful respiratory function monitoring and early treatment with specific drugs (i.e., remdesivir, monoclonal antibodies, nirmatrelvir/ritonavir, molnupiravir), when indicated, to prevent a progression of the disease.

## Figures and Tables

**Figure 1 vaccines-10-02043-f001:**
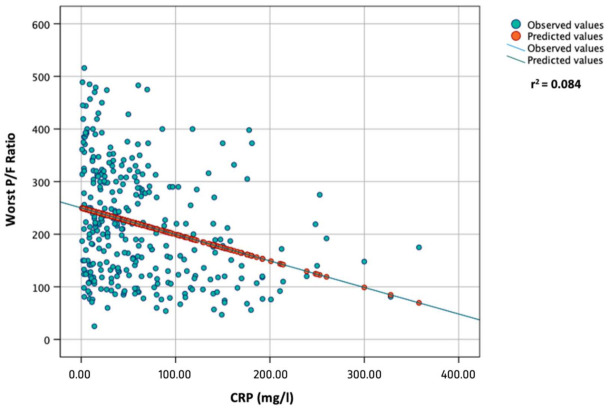
Linear regression analysis between worst P/F ratio during hospitalization (dependent) and CRP levels at admission.

**Figure 2 vaccines-10-02043-f002:**
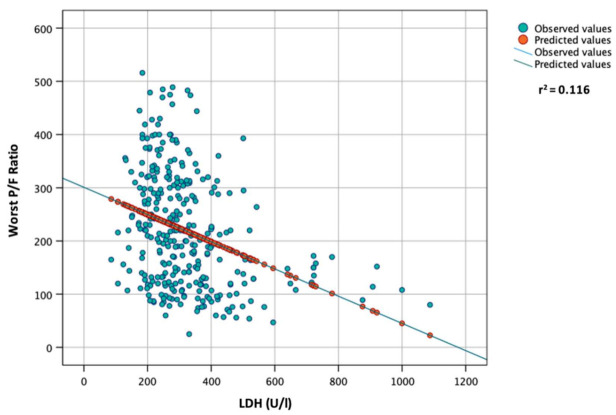
Linear regression analysis between worst P/F ratio during hospitalization (dependent) and LDH levels at admission.

**Figure 3 vaccines-10-02043-f003:**
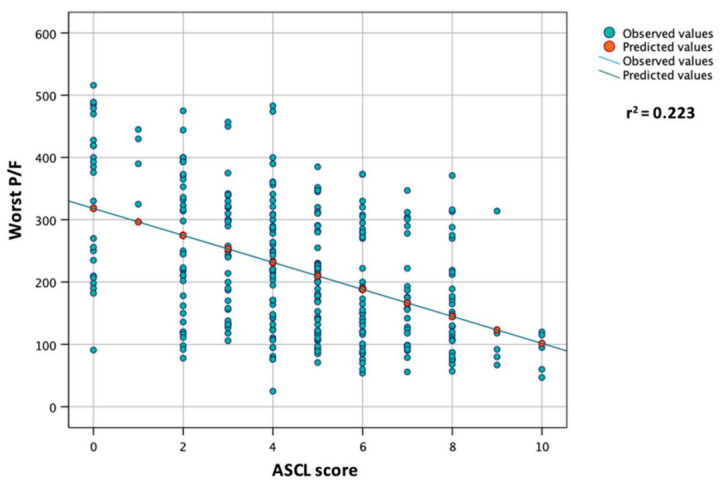
Linear regression analysis between worst P/F ratio during hospitalization (dependent) and ASCL score at admission.

**Figure 4 vaccines-10-02043-f004:**
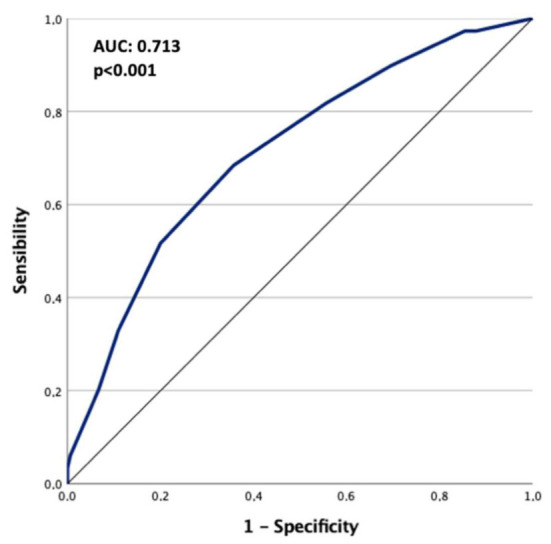
ROC curve for the diagnostic accuracy of the ASCL score in predicting P/F. Ratio deterioration below 200.

**Table 1 vaccines-10-02043-t001:** Demographic and clinical characteristics of patients hospitalized for COVID-19 and included in the study (N = 323).

Sex (M; n, %)	204 (63.2)
Age (median, IQR)	61 (49–70)
Age > 60 years (n, %)	163 (50.5)
Comorbidities (n, %)	
Cardiovascular disease	55 (17.0)
COPD	54 (16.7)
CKD	15 (4.6)
Malignancy	41 (12.7)
Cirrhosis	3 (0.9)
Diabetes	53 (16.4)
N° of comorbidities (n, %)	
0	187 (57.9)
1–2	112 (34.7)
3–5	24 (7.4)
Laboratory parameters at admission (median, IQR)	
CRP (mg/L)	41.15 (15.10–88.75)
LDH (U/L)	288 (230–369)
Lymphocyte count (cell/µL)	990 (680–1432)
Outcome (n, %)	
Worst P/F ratio < 200	153 (47.4)
ICU admission	51 (15.8)
Exitus	22 (6.8)

COPD: chronic obstructive pulmonary disease. CKD: chronic kidney disease. CRP: c-reactive protein. LDH: lactate dehydrogenase. ICU: intensive care unite

**Table 2 vaccines-10-02043-t002:** Differences in the demographic and laboratory parameters among included patients, stratified according to the presence or the absence of three unfavourable outcomes (Worst P/F ratio < 200 during hospitalization, ICU admission, death).

	Worst P/F			ICU			Death		
	<200	≥200	*p*-Value	Yes	No	*p*-Value	Yes	No	*p*-Value
Male Sex (n, %)	70.9	58.1	<0.05	78.4	61.4	<0.001	68.2	63.9	0.683
Age (median, IQR)	63 (54–72)	58 (42–67)	<0.001	65 (52–71)	60 (49–70)	0.132	78 (71–84)	60 (48–68)	<0.001
Age > 60 years (n, %)	56.4	37.7	<0.001	66.0	49.2	<0.05	90.9	49.0	<0.001
Comorbidities (n, %)									
Cardiovascular disease	18.3	15.9	0.564	17.6	16.9	0.898	50.0	14.6	<0.001
COPD	19.0	14.7	0.307	7.8	18.4	0.064	22.7	16.3	0.298
CKD	3.9	5.3	0.558	3.9	4.8	0.568	13.6	4.0	<0.05
Malignancy	14.4	11.2	0.388	7.8	13.6	0.257	27.3	11.6	<0.05
Cirrhosis	0.7	1.2	0.625	0.0	1.1	0.596	0.0	1.0	0.638
Diabetes	17.0	15.9	0.788	15.7	16.5	0.879	45.5	14.3	<0.001
N° of comorbidities (n, %)									
0	54.9	60.6	0.301	62.7	57.0	0.445	22.7	60.5	<0.001
1–2	37.9	31.8	0.247	35.3	34.6	0.919	54.4	33.2	<0.05
3–5	7.2	7.6	0.876	2.0	8.5	0.081	22.7	6.3	<0.01
Baseline CRP (mg/L; median, IQR)	60.0 (21.1–129.9)	32.0 (14.30–60.10)	<0.001	77.4 (12.0–137.0)	39.0 (16.0–75.0)	0.059	87.15 (45.40–149.0)	38.5 (15.0–80.0)	<0.001
Baseline CRP > 60 mg/L (n, %)	49.0	25.5	<0.001	52.9	33.6	<0.01	68.2	34.4	<0.01
Baseline LDH (U/L; median, IQR)	342 (256–427)	269 (211–321)	<0.001	357 (258–479)	280 (220–351)	<0.001	337 (254–479)	287 (228–360)	<0.05
Baseline LDH > 600 U/L (n,%)	10.1	0.0	<0.001	16.0	2.7	<0.001	13.6	4.1	<0.05
Baseline LDH > 300 U/L (n, %)	59.7	32.3	<0.001	62.0	42.2	<0.05	59.1	44.3	0.180
Baseline lymphocyte count (cell/µL; median, IQR)	861 (605–1220)	1100 (720–1550)	<0.001	880 (520–1150)	1000 (690–1450)	<0.05	670 (430–920)	1000 (690–1440)	<0.01
Baseline lymphocyte count < 1000 cell/µL (n, %)	62.2	43.4	<0.001	64.7	49.8	0.051	76.2	50.5	<0.05

ICU: intensive care unit. IQR: interquartile range. COPD: chronic obstructive pulmonary disease. CKD: chronic kidney disease. CRP: c-reactive protein. LDH: lactate dehydrogenase.

**Table 3 vaccines-10-02043-t003:** Univariate and multivariate linear regression analysis between the worst P/F ratio during hospitalization (dependent), age, and laboratory parameters at admission.

	Univariate Analysis			Multivariate Analysis		
	*B*	95CI	*p*-Value	*B*	95CI	*p*-Value
Worst P/F ratio ^#^	-	-	-	-	-	-
Age	−2.372	−3.073 to −1.672	<0.001	−2.079	−2.724 to −1.433	<0.001
CRP	−0.504	−0.690 to −0.319	<0.001	−0.323	−0.497 to −0.149	<0.001
LDH	−0.256	−0.335 to −0.177	<0.001	−0.205	−0.279 to −0.130	<0.001
Lymphocyte	0.000	−0.005 to +0.006	0.862	-	-	-

^#^ Worst P/F ratio was set as dependant variable. *B*: B coefficient. 95CI: 95% confidence intervals. CRP: c-reactive protein. LDH: lactate dehydrogenase.

**Table 4 vaccines-10-02043-t004:** Univariate and multivariate logistic regression analysis for worst P/F ratio < 200.

	Univariate Analysis			Multivariate Analysis		
	OR	95CI	*p*-Value	aOR	95CI	*p*-Value
**Worst P/F ratio < 200**						
Male sex	1.75	1.10 to 2.80	<0.05	1.73	1.03 to 2.91	<0.05
Age > 60 years	2.14	1.36 to 3.56	<0.001	1.80	1.10 to 2.94	<0.05
1–2 comorbidities	1.31	0.83 to 2.08	0.247	-	-	-
3–5 comorbidities	0.94	0.41 to 2.15	0.936	-	-	-
CRP > 60 mg/L	2.81	1.75 to 4.52	<0.001	2.33	1.37 to 3.94	<0.01
LDH > 300 U/L	3.11	1.95 to 4.93	<0.001	2.47	1.50 to 4.06	<0.001
Lymphocyte < 1000 cell/µL	2.14	1.36 to 3.37	<0.001	1.38	0.83 to 2.29	0.209
**ICU admission**						
Male sex	2.28	1.12 to 4.65	<0.05	2.31	1.08 to 4.92	<0.05
Age > 60 years	2.00	1.06 to 3.77	<0.05	1.66	0.86 to 3.21	0.130
1–2 comorbidities	1.03	0.55 to 1.93	0.919	-	-	-
3–5 comorbidities	0.22	0.03 to 1.64	0.214	-	-	-
CRP > 60 mg/L	2.22	1.21 to 4.08	0.01	2.00	1.03 to 3.86	<0.05
LDH > 300 U/L	2.23	1.20 to 4.16	<0.05	1.74	0.89 to 3.41	0.107
Lymphocyte < 1000 cell/µL	1.85	0.99 to 3.44	0.054	1.18	0.60 to 2.33	0.628
**Death**						
Male sex	1.21	0.48 to 3.07	0.683	0.93	0.33 to 2.60	0.885
Age > 60 years	10.42	2.39 to 45,39	<0.01	8.65	1.86 to 40.33	<0.01
1–2 comorbidities	2.41	1.01 to 5.77	<0.05	2.85	0.92 to 8.87	0.07
3–5 comorbidities	4.36	1.45 to 13.11	<0.01	8.17	1.72 to 38.71	<0.01
CRP > 60 mg/L	4.09	1.62 to 10.37	<0.01	5.45	1.82 to 16.34	<0.01
LDH > 300 U/L	1.81	0.75 to 4.38	0.185	1.02	0.36 to 2.90	0.969
Lymphocyte < 1000 cell/µL	3.13	1.12 to 8.78	<0.05	2.20	0.71 to 6.78	0.169

Bold: Dependant variables. OR: odds ratio. 95CI: 95% confidence intervals. aOR: adjusted odds ratio. CRP: c-reactive protein. LDH: lactate dehydrogenase.

**Table 5 vaccines-10-02043-t005:** The ASCL score, based on age, sex, CRP at hospital admission, and LDH at hospital admission.

Parameter	Points
Age	
<50 years	0
50–59 years	1
60–69 years	2
70–79 years	3
≥80 years	4
Sex	
Female	0
Male	2
CRP	
≤60 mg/L	0
>60 mg/L	2
LDH	
≤300 U/L	0
>300 U/L	2

OR: odds ratio. 95CI: 95% confidence intervals. aOR: adjusted odds ratio. CRP: c-reactive protein. LDH: lactate dehydrogenase.

**Table 6 vaccines-10-02043-t006:** Logistic regression analysis for P/F ratio < 200 during hospitalization according to the ASCL score.

	P/F Ratio < 200 (*n* = 153)		
ASCL Score	% *	OR	95CI
0	16.7	0.20	0.07 to 0.60
1	0.0	^#^	^#^
2	29.7	0.43	0.20 to 0.90
3	34.3	0.54	0.26 to 1.13
4	37.7	0.62	0.34 to 1.14
5	49.0	1.08	0.59 to 1.97
6	65.1	2.31	1.18 to 4.52
7	73.1	3.30	1.35 to 8.09
8	67.7	2.54	1.16 to 5.59
9	80.0	4.54	0.50 to 41.04
10	100.0	^‡^	^‡^

* Raw percentage. ^#^ Incalculable: no patients with ASCL score = 1 had a P/F ratio < 200. ^‡^ Incalculable: all patients with ASCL score = 10 had a P/F ratio < 200. ASCL: age, sex, CRP, LDH. ICU: intensive care unit. OR: odds ratio. 95CI: 95% confidence intervals.

## Data Availability

The data presented in this study are available on request from the corresponding author.

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
