# Peer review of "A Simple Non-Invasive Score Based on Baseline Parameters Can Predict Outcome in Patients with COVID-19"

_vaccines, 2022, doi:10.3390/vaccines10122043_

Round 1
Reviewer 1 Report
In this paper , authors are evaluating some laboratory parameters of patients hospitalized for Covid 19 infection, in order to predict the worsening of their condition, as expressed by their death or ICU admission. The score they are suggesting is interesting.
The study is well designed and presented , the results are clear and easy to understand, discussion is also satisfying. English language needs some small corrections which I am going to underline below.
Tables need correction in formatting (tables 1 and 2 specially, the first letters are missing , lines have moved and the results are not well presented because of that, table 5 the dash in the first row)
Line 36 regardless of, not regardless to
Line 78 for inclusion, not for the inclusion
Reviewer 2 Report
This is a retrospective study on 323 patients with COVID-19 in four Italian hospitals. The purpose was to identify biomarkers associated with worst outcome as defined with P/F ratio (the ratio between arterial oxygen and fraction of inspired oxygen) < 200. The authors end up suggesting that a combination of age, sex, CRP and LDH on admission could be used for a numerical 11-point score to predict the worst outcome.
Even by the authors’ own conclusion, the score had some predictive value only at extreme ends, 0-2 points and 6-8 points, respectively. This does appear very useful for clinical practice.
The title is misleading as it states that CRP is used for the score, but in reality, also LDH is part of the score, in addition to clinical parameters. For CRP the problem is that it was measured only once, within 48 hours of admission, which is a wide range. Moreover, many patients were excluded because CRP was not measured at all.
Several biomarkers have been found to be associated with poor outcome in COVID-19 patients. These include not only elevated CRP and LDH, but also PCT, CK, AST, ALT and creatinine, in addition to lymphopenia and thrombocytopenia. Choosing two of these, CRP and LDH, for the present study does not add anything. Combining the biomarkers with well known risk factors age and male sex is just artificial for creation of a score.
The scatter diagrams of CRP and LDH as a function of P/F ratio, presented in Fig. 1 and 2, respectively, are illustrative and of some value, but the conclusion is not obvious. These parameters do not appear to be particularly useful for the prediction of poor outcome in individual cases.
In summary, the study does not add much to what is already known. The score appears artificial and not useful for clinical practice.
Round 2
Reviewer 2 Report
accept as revised